# Risk Factors for Venous Thromboembolism in Acute Promyelocytic Leukemia

**DOI:** 10.3390/cancers16244209

**Published:** 2024-12-17

**Authors:** Nikica Sabljic, Nikola Pantic, Marijana Virijevic, Jovan Rajic, Mirjana Cvetkovic, Lazar Trajkovic, Zlatko Pravdic, Zoran Bukumiric, Nada Suvajdzic Vukovic, Andrija Bogdanovic, Ana Vidovic, Milena Todorovic Balint, Jelena Bila, Danijela Lekovic, Irena Djunic, Darko Antic, Mirjana Mitrovic

**Affiliations:** 1Clinic of Hematology, University Clinical Center of Serbia, 11000 Belgrade, Serbia; 2Department of Internal Medicine, Faculty of Medicine, University of Belgrade, 11000 Belgrade, Serbia; 3Faculty of Medicine, Institute for Medical Statistics and Informatics, University of Belgrade, 11000 Belgrade, Serbia

**Keywords:** acute promyelocytic leukemia, venous thromboembolism, risk assessment models

## Abstract

Data regarding venous thromboembolism (VTE) predictors in patients with acute promyelocytic leukemia (APL) are scarce. A retrospective study was conducted on adult patients with APL treated with all-trans retinoic acid and anthracycline. Demographic and clinical–laboratory data related to VTE were collected and analyzed, along with the predictive value of two risk assessment models (RAMs). Among the 155 patients, 28 (18.1%) had VTEs. Neither the Al-Ani RAM nor the RAM predicted VTEs in our cohort. The independent risk factors for VTE were CD114 expression (*p* = 0.005, OR = 6.4 IC 95%: [1.8–23.2]) and the absence of bleeding at presentation (*p* = 0.013, OR = 0.086 IC 95%: [0.01–0.59]). To the best of our knowledge, this is the first study showing that a higher expression of CD114 increases the risk of VTE. So far, no risk assessment model has been sufficient to stratify patients with APL according to their risk of VTE.

## 1. Introduction

Acute promyelocytic leukemia (APL) is a rare type of acute myeloid leukemia characterized by specific clinical and biological behavior, as well as a unique treatment and prognosis [1]. Despite the revolution catalyzed by the introduction of the novel agents all-trans retinoic acid (ATRA) and arsenic trioxide (ATO), which achieved complete remission in >90% of cases with a 10-year survival rate of >80%, observed coagulopathy represents a major obstacle [1,2,3]. Although the most common and frightening complication is severe hemorrhage, patients with APL are also prone to thrombosis [1,2]. According to the reported data from retrospective trials, the incidence of thrombosis in APL during the post-ATRA era varies from 5% to 20% [4,5,6,7,8,9,10,11,12,13]. Both venous and arterial thromboses have been observed, with venous thromboembolism (VTE) incidence rates between 3% and 14.3% [4,5,6,7,8,9,10,12,13]. The most common types of VTE are deep vein thrombosis (DVT) and pulmonary embolism [4,5,6,7,8,9,10,12,13]. Despite the effort to identify the predictors for VTE, none of them are reliable. Across laboratory data, an increased white blood cell count (WBC), shorter prothrombin time (PT), activated partial thromboplastin time (APTT), and a lower International Society on Thrombosis and Haemostasis (ISTH) DIC score were observed to be associated with thrombotic complications [6,9,13]. Among the molecular and immunophenotypic markers, PML/RARA bcr3 type, FLT3-ITD mutation, and CD2 and CD15 expression emerged as predictors for thromboembolism [6,9,12,14]. Moreover, polymorphism in PAI-1 4G/4G has appeared as a predictor for thrombosis in patients with APL, especially for VTE [9,12,13]. Differentiation syndrome (DS), one of the most frequent causes of early death during the ATRA era, was also associated with thrombosis [14]. More recently, a few biochemical parameters, such as elevated LDH (serum lactate dehydrogenase) and indirect bilirubin, were also proposed as predictors for thrombosis [15]. Risk assessment models (RAMs) are widely used in patients with malignancies to identify those at high risk for thrombosis. More recently, two groups of authors, Al-Ani and Paterno and colleagues, designed predictive RAMs for patients with all types of leukemia, including APL, but these scores have not yet been externally validated [16,17].

Given the limited data on the overall incidence of VTE throughout the course of APL and the absence of a reliable RAM, we conducted a retrospective trial with the aims of identifying the incidence of VTE, determining the predictive value of demographic and clinical–laboratory parameters, and analyzing the effectiveness of Al-Ani’s RAM and the RAM proposed by Paterno et al. [17].

## 2. Materials and Methods

### 2.1. Patients, Diagnostic Procedures, and Treatment

During the studied period of 2004–2024, 155 consecutive patients with APL were diagnosed and treated at the Clinic of Hematology. The diagnostics comprised the morphology, cytogenetics, molecular genetics, and immunophenotyping of bone marrow (BM). The morphological diagnosis was established according to the French–American–British classification and the WHO’s 2016 revision of the classification of myeloid neoplasms and acute leukemia [18,19]. Conventional G-band karyotyping was employed for cytogenetic analysis [20]. The diagnosis was confirmed in all patients via t(15;17)(q22;q12) detection or PML/RARA rearrangements using reverse transcriptase polymerase chain reaction (RT-PCR) [21]. Immunophenotyping was performed at diagnosis using direct multicolor immunofluorescence applied to the whole bone marrow specimen with a wide panel of monoclonal antibodies (CD2, CD3, cCD3, CD7, CD11a, CD11b, CD11c, CD13, CD15, CD16, CD19, CD33, CD34, CD38, CD45, CD56, CD64, CD114, CD117, HLA-DR, and MPO) [22]. For the expression status of the markers, positivity was defined as expression in ≥20% of cells. All patients were treated with ATRA plus idarubicin during induction (AIDA regimen), followed by three ATRA plus polychemotherapy consolidation cycles according to PETHEMA APL 99 (from September 2004 to August 2010), PETHEMA APL 2005 (from September 2010 to May 2013), and PETHEMA APL 2012 protocols (from June 2013) [22,23,24]. Patients with WBC > 5 × 10^9^/L at presentation received differentiation syndrome (DS) prophylaxis with corticosteroids from day 1 to day 15 [23,24,25]. The diagnosis of DS was established according to the presence of at least 2 of the following signs and symptoms: dyspnea, unexplained fever, weight gain greater than 5 kg, unexplained hypotension, acute renal failure, and chest radiograph demonstrating pulmonary infiltrates or pleuro-pericardial effusion [26]. Patients with confirmed DS were treated with dexamethasone, and ATRA was withheld until stabilization. Platelet concentrate transfusions were used to obtain platelet count > 30–50 × 10^9^/L, while cryoprecipitate was used to obtain fibrinogen level > 1–1.5 g/L. Fresh frozen plasma was transfused in case of INR < 1.5. These supportive measures were applied until coagulopathy was corrected, previously defined as the normalization of fibrinogen and PT and bleeding cessation [24,25,27]. Antifibrinolytics and prophylactic anticoagulant therapy were not used. For patients who were already on anticoagulants due to other medical conditions, anticoagulation was continued with low-molecular-weight heparin (LMWH) according to platelet count and hemorrhagic syndrome. Central venous catheters (CVCs) were not inserted before coagulopathy had ceased during induction but were commonly placed during cycles of consolidation. All patients were followed up from diagnosis until death or 6 months after diagnosis.

### 2.2. Data Collection

Data collection included demographic features (age and gender), body mass index (BMI), the presence of comorbidities (hypertension, diabetes mellitus, atrial fibrillation, a history of previous thrombosis, or other comorbidities), Eastern Cooperative Oncology Group performance status (ECOG PS), baseline laboratory findings (complete blood count, fibrinogen, PT, APTT, and D-dimer), immunophenotype, cytogenetics (t(15;17), normal karyotype, or additional chromosomal abnormalities), molecular genetics (the type of PML/RARA fusion), bleeding at presentation, and the presence of DS. We evaluated the presence of disseminated intravascular coagulation (DIC) before therapy was initiated. The DIC diagnosis was established according to the recommendations of the International Society on Thrombosis and Haemostasis (ISTH). Patients with an ISTH DIC score ≥ 5 were considered to have overt DIC [28]. The Al-Ani risk score was calculated in all patients according to the following formula: history of previous VTE (3 points), acute lymphoblastic leukemia (2 points), and platelet count > 50 × 10^9^/L (1 point). According to the score, the patients were assigned to low- (0–2 points) or high-risk (≥3 points) categories [16]. The Paterno risk score was calculated according to the formula comorbidities (1 point), platelet count > 50 × 10^9^/L (1 point), and history of previous VTE (1 point). According to the score, patients with a score of 0–1 were assigned to the low-risk group, while patients with a score ≥ 2 were assigned to the high-risk group [17].

### 2.3. Thrombotic Events and Therapy

The primary outcome was the occurrence of symptomatic imaging-confirmed VTE, including lower-limb DVT, pulmonary embolism (PE), thrombosis of unusual sites (upper extremity deep venous thrombosis, cerebral vein thrombosis, and Budd–Chiari syndrome), and symptomatic CVC-related thrombosis. DVT- or CVC-related thrombosis diagnosis required the compression ultrasound evidence of a thrombus. Acute PE was defined as the presence of filling defects on computed tomography pulmonary angiography. Budd–Chiari syndrome (BCS) and cerebral vein thrombosis were diagnosed using computed tomography. All patients with VTE were treated with low-molecular-weight heparin (LMWH) for at least three months. The LMWH dose was adjusted according to the platelet count. If the platelet count was ≥50 × 10^9^/L, a full therapeutic dose was administered; if the platelet count was between 30 and 50 × 10^9^/L, half of the dose was applied; and, if the platelet count was <30 × 109/L, LMWH was omitted [29].

### 2.4. Statistical Analyses

Statistical analysis was performed using IBM SPSS statistical software (SPSS for Windows, release 25.0, SPSS, Chicago, IL, USA). Categorical variables are presented as absolute or relative frequencies. For the comparison of categorical variables, either the chi-squared test or Fisher’s exact test was employed. The chi-squared test was applied for larger sample sizes with expected frequencies greater than 5, while Fisher’s test was used when the expected frequencies in any cell were below 5. The Kolmogorov–Smirnov test and histogram were used to assess the normality of data distribution. Continuous variables with normal distribution are shown as mean and standard deviation (mean ± SD), while variables not complying with normal distribution are presented as median and range. The *t*-test or Mann–Whitney U test was used to compare groups for continuous variables. The significance level was set at 0.05. Univariate and multivariate logistic regression analyses were used to calculate and validate the risk factors for thrombosis development. Variables with a *p* < 0.05 in the univariate model were included in the multivariate logistic regression analysis.

## 3. Results

Among the 155 patients with a median age of 44 years (ranging from 18 to 78 years), 81 (52%) were females. At presentation, mild hemorrhagic syndrome was noted in 131/155 (84%) patients, severe hemorrhage was observed in 12/155 (8%) patients, and bleeding was absent in 12/155 (8%) patients. The diagnosis of DS was established in 52/155 (33.5%) subjects. Early death occurred in 38/155 (24.5%) subjects within 30 days of diagnosis. Comorbidities were observed in 48/143 (33.6%) patients, with hypertension being the most common (35/143, 24.5%). A history of previous thromboses was noted in 8/143 (5.6%) patients, of whom only two (1.4%) had a prior diagnosis of VTE. Six patients were already on long-term anticoagulation therapy: five due to atrial fibrillation and one due to the presence of a mechanical aortic valve. Two patients with atrial fibrillation died on day 1 and day 2, respectively, before anticoagulation with LMWH could be continued, while, in a patient with a mechanical valve, the switch from vitamin K antagonist to LMWH occurred.

### 3.1. VTE Events

We observed 28 cases of VTE, so the overall incidence of VTE across the whole course of the disease was 18.1%. The female-to-male ratio was 1. VTE encompassed CVC-related thrombosis in twelve (42.8%), DVT in six (21.4%), PE in one patient (3.6%), and unusual sites in nine (32.1%) cases. Among the nine VTE cases of unusual sites, six involved DVT of the upper extremity, two were thromboses of the sagittal sinus, and one was Budd–Chiari syndrome. In four cases (14.2%), VTE was present at diagnosis prior to leukemia treatment, while eleven (39.2%) and thirteen (46.4%) occurred during induction and consolidations, respectively. Among the VTEs observed at presentation, three were DVT and one was PE. Among the CVC-related thromboses, the majority, 9/12 (75%), were diagnosed during consolidations, while only two (3/12, 25%) occurred during induction. The most frequent type of VTE during induction was thrombosis of unusual sites (7/11, 63.6%). The type of VTE and its occurrence during the course of the disease are presented in Figure 1. Only one patient with a VTE at presentation died within 30 days, but this was due to an intracranial hemorrhage on day 1. VTE did not have an influence on the early death rate (*p* = 0.23).

### 3.2. Predictors for VTE

The clinical and laboratory features and comparison of patients with APL with and without VTEs are shown in Table 1 and Table 2. The patients with VTEs had significantly shorter PT (72.5% vs. 63.5%, *p* = 0.01), higher expression of the CD114 molecule (70.4% vs. 41.7%, *p* = 0.007), and lower ISTH DIC scores (5 vs. 6, *p* = 0.033). The absence of bleeding at presentation was also associated with VTE (17.9% vs. 5.5%, *p* = 0.43). Moreover, those patients diagnosed with VTEs during the course of the disease more often reported previous thrombotic events (14.3% vs. 3.1%, *p* = 0.042). There was no significant difference between the two groups regarding other parameters. The most significant risk factors for VTE occurrence in multivariate Cox’s proportional regression were CD114 expression (OR = 6.4, 95% CI: [1.8–23.2]) and the absence of bleeding at presentation (OR = 0.086, 95% CI: [0.01–0.59]), as shown in Figure 2, while the history of previous thromboses and DIC showed no significance. None of the RAMs were predictive in our cohort. The C statistics values for Al-Ani’s model and Paterno’s RAM were ROC = 0.514 and ROC = 0.521, respectively.

## 4. Discussion

It has been suggested that thromboembolic complications in APL are more frequent than in any other type of acute myeloid leukemia (AML). While data regarding the 6-month cumulative incidence of VTE in APL are scarce, novel data have shown a 6-month cumulative incidence of 13.8% in non-promyelocytic AML [30]. According to previous studies, the VTE incidence in APL varies between 3% and 14.3% [4,5,6,7,8,9,10,12,13,14,15]. The majority of those studies were focused only on presentation and the induction period [4,5,6,7,8,9,10,12]. A study by Ben Salah et al. that investigated the thromboembolic complications, both arterial and venous, across the whole treatment period, from induction to consolidation, reported a VTE incidence of 5.5% [14]. Of note, in this retrospective trial, the authors considered PE as arterial thrombosis, which could further influence the incidence of VTE. In our group, we registered an overall VTE incidence of 18.1%. A higher incidence of VTE among our group of patients could consequently be due to focusing not only on VTE occurring at presentation and during induction but also on VTE arising during consolidation. In our cohort, all the patients were treated with the AIDA regimen without ATO. The effects of ATO on potential thrombotic complications are not clear. There are not many data on the incidence of thrombosis in the ATO era, nor direct comparisons between the AIDA regimen and ATO-based regimens. However, in several studies in which the patients were treated with ATO, the incidence of VTE was similar (7.9% and 13.6%) to that described in the pre-ATO era [8,12].

Our results suggest that all VTE events could be divided into two groups regarding their localization. The predominantly observed types of VTE during induction and at presentation were DVT, PTE, and other types of VTE. This is likely driven by APL itself, as well as the specific procoagulant activity of APL cells and the endothelium [31]. On the other hand, CVC-related thrombosis was the most frequent type identified during consolidation. While CVC-related thrombosis accounts for 75% of all the VTEs in AML, the real incidence of CVC-related thromboses in APL is unclear [17,32]. We reported that 42.8% of all the VTEs in our cohort were CVC-related thromboses, most of which occurred during consolidation (9/12, 75%). Only two trials analyzed CVC-related thrombosis in APL, both during induction, and showed that 50% of all the thromboses and 30% of the VTEs were CVC-related [8,9]. It should be mentioned that the current guidelines advocate against CVC insertion in APL during induction until coagulopathy cessation. Furthermore, there are several patients assigned to the low-risk APL group who could be treated with chemotherapy-free regimens in whom the insertion of a CVC is not necessary. All these factors could influence the underestimated incidence of CVC-related thrombosis in APL.

We described the association of different clinical–laboratory parameters with VTE. In our cohort, lower PT was predictive for VTE, which is in line with some previous reports from the literature [9,14]. Although thromboses are described in conditions with DIC, in our group, those patients with thromboses had lower median ISTH DIC scores in comparison with the patients without VTE. Similar results have been previously reported [9]. Conversely, in studies investigating hemorrhagic complications in APL, higher DIC scores were associated with severe bleeding [33,34]. It is possible that such patients with higher DIC scores exhibited very pronounced consumption of clotting factors, resulting in a hypocoagulable state that increases their susceptibility to bleeding. In our study group, we also showed that patients with no hemorrhage at presentation more often developed VTEs. While previous papers observed that patients with APL could present with both bleeding and thrombosis, none of them showed a significant association with the lack of bleeding and VTE [4,8].

To the best of our knowledge, this is the first trial in which CD114 expression emerged as the most important predictor for VTE in patients with APL. Higher CD114 molecule expression led to a 6-fold higher risk of VTE. CD114 is a granulocyte colony-stimulating factor (G-CSF) receptor and can be found not only on abnormal leukemic cells but also on healthy granulocyte precursors. It has already been postulated that G-CSF can induce a hypercoagulable state via tissue factor stimulation, an increment in platelet aggregation, and an inflammatory state [35,36,37]. Moreover, some reports have suggested the presence of G-CSF-induced acute arterial thrombosis in healthy individuals, as well as in patients with coronary artery disease [38,39]. While G-CSF use in patients with APL is generally not recommended, the use of corticosteroids, which can increase the levels of endogenous G-CSF, is widespread in order to prevent or treat DS [40].

Thrombocytopenia and DIC are usually present in patients with AML, and the challenging use of anticoagulants due to bleeding risk causes patients with AML to generally be underrepresented in trials dealing with risk assessment models for VTE. Major efforts have been undertaken to design a few RAMs for VTE in patients with AML; however, some studies have excluded patients with APL [17,32]. More recently, a new RAM for patients with all types of acute leukemia, including 41 with APL, was developed by Al-Ani and colleagues [16]. Its score is derived from the history of previous VTE, type of acute leukemia, and platelet count and further divides the patient into low- and high-risk groups [16]. Additionally, a new score for patients with AML was proposed by Paterno et al. The score comprises three parameters: comorbidities, platelet count, and previous VTE. According to the final score, the patients are separated into low- and high-risk categories [17]. Even though the authors did not clarify whether patients with APL were included or excluded, we aimed to analyze the efficacy of the score on our group. Our analysis failed to show the significance of both RAMs in our cohort.

This study has several limitations, such as a small sample size and its retrospective single-center nature, which could restrain the ability to identify all the predictors for VTE. Moreover, we analyzed a cohort of patients treated with chemotherapy and ATRA without ATO. We still believe that the results presented here could be useful in several ways. We observed the whole course of APL and provided information about thrombotic complications during consolidation. We also emphasized the importance of CVC-related thromboses in APL and identified some potential new predictors for VTE, namely CD114 and the absence of bleeding at presentation. However, severe hemorrhages and hemorrhagic early death still present the greatest obstacle in APL treatment, reaching almost 16% in real-world clinical practice [41]. On the contrary, we showed that none of the VTEs contributed to early death. Thus, further studies are needed to evaluate the risks and benefits of the eventual introduction of thromboprophylaxis in carefully identified high-risk patients.

## 5. Conclusions

Our study suggests that VTE is not only a common problem in APL at presentation and during induction but should also be considered in patients during consolidations. In addition, we showed that VTE is associated with different laboratory and clinical parameters, which could be useful in identifying patients at higher risk for VTEs. Additionally, we could not confirm the efficacy of the Al-Ani score and the new RAM proposed by Paterno et al. [17] Additional studies are needed to create more accurate RAMs for APL. We aim to shed new light on the identification of patients at increased risk of VTEs and potentially provide more supportive therapy.

## Figures and Tables

**Figure 1 cancers-16-04209-f001:**
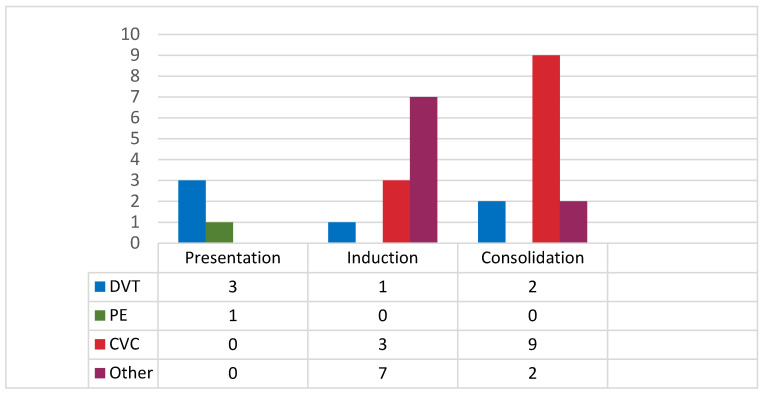
Type of VTE and its occurrence during the course of disease.

**Figure 2 cancers-16-04209-f002:**
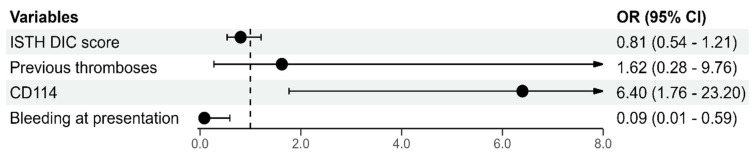
Multivariate analyses of predictors for VTE in patients with APL.

**Table 1 cancers-16-04209-t001:** Comparison of clinical and laboratory data between patients with and without VTEs.

Parameter		Patients Without VTE (*n* = 127)	Patients with VTE (*n* = 28)	*p* Value
Age (years), mean ± SD		43.76 ± 16.04	48.79 ± 16.34	0.137 ^a^
Female gender (females), N (%)		68 (53.5%)	13 (46.4%)	0.495 ^b^
BMI > 30 kg/m^2^		22 (17.3%)	8 (28.6%)	0.220 ^b^
ECOG PS, N (%)	0	13 (10.2%)	2 (7.1%)	0.635 ^c^
	1	52 (40.9%)	14 (50%)	
	2	28 (22%)	7 (25%)	
	3	28 (22%)	3 (10.7%)	
	4	6 (4.7%)	2 (7.1%)	
Comorbidities, N (%)	Hypertension	26 (20.5%)	9 (32.1%)	0.235 ^b^
	Diabetes mellitus	8 (6.3%)	3 (10.7%)	0.435 ^d^
	Atrial fibrillation	4 (3.1%)	1 (3.6%)	0.987 ^d^
	Other	15 (11.8%)	5 (17.8%)	0.537 ^d^
History of previous thromboses, N (%)		4 (3.1%)	4 (14.3%)	**0.042** ^d^
History of previous VTE, N (%)		1 (0.8%)	1 (3.6%)	0.478 ^d^
Bleeding at presentation, N (%)	No	7 (5.5%)	5 (17.9%)	**0.043** ^d^
	Yes	120 (94.5%)	23 (82.1%)	
Differentiation syndrome, N (%)		42 (33.1%)	10 (35.7%)	0.789 ^b^
WBC (×10^9^/L), median (range)		4.4 (0.4–260.4)	2.7 (0.3–118.6)	0.103 ^c^
Platelet count (×10^9^/L), median (range)		27 (0–279)	30.5 (4–194)	0.750 ^c^
Hemoglobin (g/L), median (range)		96 (39–160)	103 (70–144)	0.166 ^c^
Fibrinogen (g/L), median (range)		2.5 (0.4–6.9)	3.48 (1.0–6.1)	0.054 ^c^
PT (%), median (range)		63.5 (21–124)	72.5 (38–137)	**0.010** ^c^
APTT (s), median (range)		25.7 (20.2–35.7)	26 (15.9–44.2)	0.376 ^c^
D dimer (mg/L), median (range)		26.2 (1.88–233)	25.6 (0.7–250)	0.749 ^c^
ISTH DIC score, median (range)		6 (2–8)	5 (1–7)	**0.033** ^c^
Peripheral blasts (%), median (range)		7 (0–97)	1 (0–83)	0.149 ^c^
PMC in peripheral blood (%), median (range)		30 (0–94)	18.5 (0–77)	0.187 ^c^
Cytogenetics, N (%)	t(15;17)	98 (79.7%)	22 (78.6%)	0.577 ^d^
	Normal karyotype	21 (17.1%)	4 (14.3%)	
	Additional abnormalities	4 (3.3%)	2 (7.1%)	
*PML/RARA* type, N (%)	bcr1	44 (61.1%)	11 (55%)	0.648 ^d^
	bcr2	3 (4.2%)	0 (0%)	
	bcr3	25 (34.7%)	9 (45%)	
Sanz risk score, N (%)	Low	29 (22.8%)	7 (25%)	0.180 ^c^
	Intermediate	48 (37.8%)	15 (53.6%)	
	High	50 (39.4%)	6 (21.4%)	
Al-Ani score, N (%)	Low risk (0–2)	117 (99.2%)	26 (96.3%)	0.339 ^d^
	High risk (≥3)	1 (0.8%)	1 (3.7%)	
Paterno score, N (%)	Low risk (0–1)	109 (93.2%)	24 (88.9%)	0.432 ^d^
	High risk (≥2)	8 (6.8%)	3 (11.1%)	

Abbreviations: APTT, activated thromboplastin time; BMI, body mass index; ISTH, International Society on Thrombosis and Haemostasis; PMCs, promyelocytes; PT, prothrombin time; WBC, white blood cell count. ^a^, *t*-test; ^b^, chi-squared test; ^c^, Mann–Whitney test; ^d^, Fisher’s exact test. In bold format we presented *p* values of statistical significance.

**Table 2 cancers-16-04209-t002:** Comparison of immunophenotype between patients with and without VTEs.

Parameter	Patients Without VTE (*n* = 111)	Patients with VTE (*n* = 23)	*p* Value
HLA-DR	8 (6.7%)	3 (11.1%)	0.425 ^a^
MPO	102 (95.3%)	25 (96.2%)	1.000 ^b^
CD2	27 (22.5%)	7 (26.9%)	0.629 ^a^
cCD3	9 (9.1%)	3 (11.5%)	0.713 ^b^
CD7	1 (0.8%)	0 (0%)	1.000 ^b^
CD13	120 (100%)	27 (100%)	1.000 ^b^
CD15	12 (10.1%)	1 (3.7%)	0.463 ^b^
CD19	6 (5%)	2 (7.4%)	0.639 ^b^
CD33	117 (97.5%)	27 (100%)	1.000 ^b^
CD34	34 (28.3%)	8 (29.6%)	0.893 ^b^
CD38	36 (40%)	8 (32%)	0.467 ^a^
CD56	12 (12%)	4 (15.4%)	0.741 ^b^
CD64	89 (74.8%)	20 (74.1%)	0.938 ^a^
CD114	50 (41.7%)	19 (70.4%)	**0.007** ^a^
CD117	113 (94.2%)	26 (96.3%)	1.000 ^b^

^a^, Chi-squared test; ^b^, Fisher’s exact test. In bold format we presented *p* value of statistical significance.

## Data Availability

The datasets used and analyzed during the current study are available from the corresponding author upon reasonable request.

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
