# Peer review of "Risk Factors for Venous Thromboembolism in Acute Promyelocytic Leukemia"

_cancers, 2024, doi:10.3390/cancers16244209_

Round 1
Reviewer 1 Report
Comments and Suggestions for Authors
The authors sought to identify potential risk factors for venous thromboembolism (VTE) in acute promyelocytic leukemia (APL) by comparing the demographic features and clinical laboratory results from APL patients with and without VTE. All patients were treated with all-trans retinoic acid (ATRA), and anthracycline. The authors claimed that the high expression of CD114 and the absence of bleeding at presentation in patients with APL correlate with VTE. The following are the major comments:
- Multiple statistical tests were performed on the data presented in Tables 1 and 2, but it is not clear what exact test was used for each comparison, preventing a thorough examination of the results.
- For multiple comparisons in Tables, if applying statistical corrections to control the false discovery rate, such as the Bonferroni correction or the Benjamini-Hochberg (BH) procedure, is there still significance for CD114, a history of previous thromboses, and the absence of bleeding at presentation?
- For immunophenotyping results, since higher expression of CD114 is a key finding in this manuscript, it is recommended to provide detailed explanations and additional information. For instance, does the data presented in Table 2 represent a single immunophenotyping result for each patient?
- The results of the multivariate Cox proportional regression analysis for “CD114 expression” (p = 0.005) and “absence of bleeding at presentation” (p = 0.013) appear inconsistent with the findings presented in Tables 1 and 2. Clarification is requested.
- The data presented in Tables 1 and 2 suggest that CD114 levels, a history of previous thromboses, and the absence of bleeding at presentation are associated with VTE in APL. Developing a new risk assessment model incorporating these three factors could be valuable. Additionally, while “CD114 expression” (p = 0.005) and “absence of bleeding at presentation” were discussed in the manuscript, the authors did not address the history of previous thromboses. Clarification on this omission is requested.
- VTE was classified into four groups: DVT, PTE, CVC, and Other in the manuscript (PTE was inconsistently referred to as PE in the main text; terminology should be kept consistent throughout the manuscript). Is it possible to perform separate comparisons of each subgroup of patients with those without VTE? Similarly, is it possible to compare patients with or without VTE at different stages of APL? If so, what results might you obtain?
- For 95% confidence intervals, it is recommended to replace all instances of 'IC 95%' with '95% CI.' Additionally, it is recommended to separate the OR value and 95% CI with a comma throughout the text.
1. Articles ('the') and commas should be removed/added where necessary, such as:
· Line 14: "the" before "VTE" should be removed.
· Line 18: "To best of our knowledge" should be revised to "To the best of our knowledge."
· Line 26: "2006 and 2024 in Clinic of" should be revised to "2006 and 2024 in the Clinic of."
· Line 54: "in APL in post-ATRA era" should be revised to "in the APL in the post-ATRA era."
· Line 163: "patients with atrial fibrillation died on day 1 and day 2, respectively before the anticoagulation with LMWH was continued" should be revised to "patients with atrial fibrillation died on day 1 and day 2, respectively, before anticoagulation with LMWH could be continued."
2. There are some other typos and errors in the manuscript, which should be carefully examined and fixed, such as
· Line 15 and 25: “RAM” should be revised to "RAMs” to indicate that there are two models.
· Line 15: “Nor Al-Ani RAM nor RAM” should be revised to “Neither the Al-Ani RAM nor the RAM”.
· Line 23: “to identify” should be revised to “To identify”.
· Line 33: “Our analyses showed nor Al-Ani RAM or RAM proposed by” should be revised to “Our analyses showed neither Al-Ani RAM nor RAM proposed by”.
· Line 88-89: The closing bracket in the monoclonal antibodies list "[20)" should be corrected to "[20]".
· Lines 93, 100, 122, 125, 138-140: “ x 109/L” should be “ x10⁹/L".
3. Some sentences in the manuscript could be considered for reorganization, such as:
· Line 157-158: “Early death within 30 days from diagnosis was experienced by 38/155 (24.5%) subjects" could be considered to revise to “Early death occurred in 38/155 (24.5%) subjects within 30 days of diagnosis”.
· Line 159-161: "History of previous thromboses had 8/143 patients (5.6%) and among them only 2 (1.4%) previously were diagnosed with VTE" could be considered to revise to "A history of previous thromboses was noted in 8/143 (5.6%) patients, of whom only 2 (1.4%) had a prior diagnosis of VTE."
Author Response
1. Multiple statistical tests were performed on the data presented in Tables 1 and 2, but it is not clear what exact test was used for each comparison, preventing a thorough examination of the results.
Response: Thank you for suggestion. We added precise information about the statistical tests used for each individual analysis in Tables 1 and 2. They are labeled as follows: a, T-test; b, Chi-square test; c, Mann Whitney test; d, Fisher exact test and added bellow Table 1 and 2.
2. For multiple comparisons in Tables, if applying statistical corrections to control the false discovery rate, such as the Bonferroni correction or the Benjamini-Hochberg (BH) procedure, is there still significance for CD114, a history of previous thromboses, and the absence of bleeding at presentation?
Response: Thank you for this very important suggestion. We considered applying statistical corrections for multiple comparisons, such as Bonferroni, Holm, FDR or Benjamini-Hochberg (BH) procedures, but decided not to do so. Our approach is based on the priority of maintaining sensitivity in detecting potentially significant findings in this specific patient population, taking into account the number of outcomes, sample size and nature of the study. We believe that the corrections could lead to overly conservative results, which would increase the risk of missing important findings.
3. For immunophenotyping results, since higher expression of CD114 is a key finding in this manuscript, it is recommended to provide detailed explanations and additional information. For instance, does the data presented in Table 2 represent a single immunophenotyping result for each patient?
Response: Immunophenotype analysis was obtained only once in every single patient, at diagnosis. Wide panel of monoclonal antibodies (CD2, CD3, cCD3, CD7, CD11a, CD11b, CD11c, CD13, CD15, CD16, CD19, CD33, CD34, CD38, CD45, CD56, CD64, CD114, CD117, HLA-DR, MPO) was analyzed. For the expression status of markers, positivity was defined as expression in ≥ 20% of cells. We added the sentence in the section Materials and Methods (Paragraph 89-90). The Table 2 represent the proportion of patients with the expression (positive) of each specific marker.
4. The results of the multivariate Cox proportional regression analysis for “CD114 expression” (p = 0.005) and “absence of bleeding at presentation” (p = 0.013) appear inconsistent with the findings presented in Tables 1 and 2. Clarification is requested.
Response: Thank you for your observation. Table 1 and Table 2 represent only results of the univariate model, while results of multivariate model are shown on Figure 2. Since Figure 2 doesn’t show the p value, if it is of interest we can remove p values from the text to avoid creating additional confusion (Paragraph 192 and 193).
5. The data presented in Tables 1 and 2 suggest that CD114 levels, a history of previous thromboses, and the absence of bleeding at presentation are associated with VTE in APL. Developing a new risk assessment model incorporating these three factors could be valuable. Additionally, while “CD114 expression” (p = 0.005) and “absence of bleeding at presentation” were discussed in the manuscript, the authors did not address the history of previous thromboses. Clarification on this omission is requested.
Response: Thank you for this observation. The aim of our study was to identify some clinical-laboratory parameters predictive for VTE. However, bearing in mind the small sample size and small number of events of interest, we decided not to preform further analysis and develop a new RAM. In the section Results (Paragraph 194), we added that neither DIC or history of previous thromboses were predictive in multivariate model. Further in the text (Discussion) we didn’t address the history of previous thromboses in order to be more focused and concise. Also, in the previous studies regarding thrombosis in APL, none of them analyzed this parameter. The history of previous thrombosis was considered only in the studies focused on creating RAMs (Al Ani and Paterno).
6. VTE was classified into four groups: DVT, PTE, CVC, and Other in the manuscript (PTE was inconsistently referred to as PE in the main text; terminology should be kept consistent throughout the manuscript). Is it possible to perform separate comparisons of each subgroup of patients with those without VTE? Similarly, is it possible to compare patients with or without VTE at different stages of APL? If so, what results might you obtain?
Response: Thank you for this comment. We made a correction in the Figure 1, changing PTE in PE in order to keep the consistency of terminology. We agree that it would be interesting to compare patients with specific type of VTE with those without VTE. However, we believe that the small sample size and number of VTEs restrains for obtaining such analysis. The same is for the comparison of patients with and without VTE regarding the stage of APL.
7. For 95% confidence intervals, it is recommended to replace all instances of 'IC 95%' with '95% CI.' Additionally, it is recommended to separate the OR value and 95% CI with a comma throughout the text.
Response: Thank you for your suggestion. We replaced “IC 95%” with “95% CI” and added coma to separate “OR” and “95% CI” (Paragraphs 192 and 193).
We had also carefully read your suggestion regarding the quality of English and made correction in the main text according to yours advises. All corrections were highlighted in the text.
Reviewer 2 Report
Comments and Suggestions for Authors
I have read with great interest the manuscript entitled "RISK FACTORS FOR VENOUS THROMBOEMBOLISM IN ACUTE PROMYELOCYTIC LEUKEMIA". It is an interesting study that points out that CD114 expression is correlated with the incidence of VTE in APL. However, at this point, the article requires major revisions.
The study sample is small, as only 28 patients were included, and the investigation is unicentric and retrospective in nature. The use of arsenic trioxide was not considered - it is not available in the frontline of APL treatment in Serbia?
You did not consider other variables that could influence the risk of thrombosis, such as WBC at diagnosis, neutrophil to lymphocyte ratio, platelet to lymphocyte ratio, levels of acute or chronic inflammation markers, etc.
How about the comorbidities of these patients? Dyslipidemia, chronic kidney disease, hypertension, diabetes, etc. increase the risk of thrombotic events.
Were there any associations with obesity classes? You only examined the impact of BMI values.
The references are too old, half of them were published earlier than 2018. You need to add more titles from recent years.
Also, there seems to be a high degree of similarity between this paper and some of your previous publications:
https://www.thrombosisresearch.com/article/S0049-3848(14)00635-5/abstract
https://www.thieme-connect.de/products/ejournals/abstract/10.1055/a-2238-4782
Can you please explain the novelty of this study in comparison with the 2015 publication?
Author Response
1. The study sample is small, as only 28 patients were included, and the investigation is unicentric and retrospective in nature. The use of arsenic trioxide was not considered - it is not available in the frontline of APL treatment in Serbia?
Response: Thank you for your observation. We analyzed data on 155 consecutive APL patients, and the number of events of interest (VTE) was 28 like you stated. As per standard methodological criteria, a minimum of 5 to 10 events per predictor studied are required for the development of a clinical prediction model, this number of VTEs allowed us to analyze at least three parameters in multivariate model. APL is a very rare disease, accounting for only 10% of all AML cases so it is not uncommon that studies concerning this pathology have a small sample size. By instance, to the best of our knowledge, our group of patients would be the second largest group among other published papers concerning APL and thrombosis (the largest group is by Montesinos et al. with 739 cases, while the second one consists of 127 patients). We agree that sample size is limitation of our study what was discussed in the limitations part of the article.
All patients were treated with AIDA regimen. Unfortunately, ATO is still not available in Serbia. However, we believe that our data, concerning VTE in ALP patients treated with chemo plus ATRA, might be helpful since many other countries worldwide lacking ATO. All of the above was discussed in the limitations part of the article
2. You did not consider other variables that could influence the risk of thrombosis, such as WBC at diagnosis, neutrophil to lymphocyte ratio, platelet to lymphocyte ratio, levels of acute or chronic inflammation markers, etc.
Response: Thank you for this comment. We analyzed several demographic, laboratory data including WBC and some clinical data. All analyzed variables are shown in the Table 1 and 2. All laboratory data were obtained at diagnosis. NLR and PLR are very interesting parameters, but we did not use it since the disease arises from the bone marrow and affects in many ways every single cell lineage which is seen by very severe changes in peripheral blood count with blast domination.
3. How about the comorbidities of these patients? Dyslipidemia, chronic kidney disease, hypertension, diabetes, etc. increase the risk of thrombotic events.
Response: Thank you for the comment. We analyzed comorbidities such as hypertension, diabetes mellitus and atrial fibrillation. All other comorbidities (e.g. CKD, dyslipidemia etc.) were classified as “other”. The “other” group comprised only of 20 patients, 15 without VTE and 5 with VTE and the sample size would be too small to be compared separately. All this is shown in Table 1. None of the comorbid state was predictive for VTE.
4. Were there any associations with obesity classes? You only examined the impact of BMI values.
Response: Thank you for your comment. We have chosen the BMI since it was the most common parameter used in the other studies. We defined obesity as BMI > 30kg/m2 as per WHO definition. This parameter had no influence on VTE as shown in Table 1. Having in mind lack of significance and small sample size we decide not to perform separate subgroups analysis (for underweight, normal, over and obese).
5. The references are too old, half of them were published earlier than 2018. You need to add more titles from recent years.
Response: Thank you for the suggestion. Unfortunately, not many research articles concerning VTE and thromboses overall in APL patients were published in past several years. The most recent data were provided by ‘Ben Salah (13) in 2022 in the form of abstract. However, papers regarding RAMs are more recent data, as well as some review articles citied in our manuscript (2, 14, 15, 29). Yet, if you have any suggestion on some specific references that could be important and are omitted in this manuscript, we are willing to add it.
6. Also, there seems to be a high degree of similarity between this paper and some of your previous publications:https://www.thrombosisresearch.com/article/S0049-3848(14)00635-5/abstract; https://www.thieme-connect.de/products/ejournals/abstract/10.1055/a-2238-4782; Can you please explain the novelty of this study in comparison with the 2015 publication?
Response: Thank you for this comment. In the previous publication by Mitrovic et al. (9), authors analyzed the group of 63 APL patients. The larger sample size allowed us to identify clinical parameters predictive for thrombosis development what was not the case in previous paper. Even more, we made external validations of two new RAMs, not existing in the time of publication of previous paper. Our recent publication (Sabljic et al.) is review article, containing no data from our group of patients.
Round 2
Reviewer 1 Report
Comments and Suggestions for Authors
1. Numerous grammatical errors and typos still exist, and the authors should carefully review the manuscript and correct them. Below are just two examples.
a. Line 221-222: The sentence “According to our results, it gives an impression that all VTE events could be divided into two groups regarding its localization' should be revised to 'According to our results, it gives the impression that all VTE events could be divided into two groups regarding their localization”.
b. Line 286: “Our study suggest VTE are frequent problem in APL not only at presentation' should be revised to 'Our study suggests that VTE is a common problem in APL not only at presentation”.
2. Considering that multiple hypothesis testing was conducted in the manuscript, the statistical results should include either the Bonferroni correction or the Benjamini-Hochberg (BH) procedure. If no significant results are found, potential reasons should be addressed/discussed in the manuscript.
3. In Tables 1 and 2, both Fisher’s exact test and the Chi-square test are used. Since the choice of test does not appear to be based solely on positive counts, the rationale for selecting Fisher’s exact test or Chi-square test for each specific comparison should be explained in the Methods section or figure legends. For similar comparisons, consistency should be maintained.
Comments on the Quality of English LanguageNumerous grammatical errors and typos still exist, and the authors should carefully review the manuscript and correct them.
Author Response
Thank you once again for your meticulous review and valuable suggestions on our manuscript.
1. We have made the English language changes as you advised and highlighted them in the manuscript.
2. Considering that multiple hypothesis testing was conducted in the manuscript, the statistical results should include either the Bonferroni correction or the Benjamini-Hochberg (BH) procedure. If no significant results are found, potential reasons should be addressed/discussed in the manuscript.
Response: Thank you for your thoughtful comments regarding multiple hypothesis testing. We would like to clarify the purpose and context of the univariate analyses conducted in our study. The multiple hypothesis testing conducted in the manuscript aimed to screen for variables through univariate analyses to be considered in the multivariable model. Corrections such as Bonferroni or Benjamini-Hochberg were not applied. Applying these corrections at this stage could exclude variables that may still be significant contributors in the adjusted multivariable context, particularly when accounting for confounding or interactions. Instead, we used a more lenient p-value threshold (e.g., p < 0.10 or p < 0.20) in the univariate analyses to ensure that important variables were not prematurely excluded. The ultimate significance of variables was determined within the multivariable model, which provides a more robust and adjusted assessment of their association with the outcome. Thus, the risk of type I errors due to multiple comparisons in the univariate stage was mitigated by the subsequent multivariable modeling process.
3. In Tables 1 and 2, both Fisher’s exact test and the Chi-square test are used. Since the choice of test does not appear to be based solely on positive counts, the rationale for selecting Fisher’s exact test or Chi-square test for each specific comparison should be explained in the Methods section or figure legends. For similar comparisons, consistency should be maintained.
Response: We added the explanation in the Section Materials and Methods (Paragraphs 148-151). Also, bellow and in the Table 1 and 2 we added the description which exact test was used for each comparison.
Reviewer 2 Report
Comments and Suggestions for Authors
The authors have answered to many of my comments but have made no effort to improve the manuscript despite the fact that it was submitted to a high-impact journal. Although I can partially agree with some of your answers, the references remain too old. There are over 12000 papers on APL in MEDLINE and many recent papers on thrombotic events in APL. I found a paper identifying biochemical predictors of thrombotic events published this year. You can also compare the AIDA regimen to ATRA plus ATO in terms of thrombosis occurrence. If you would read other studies in full length and not only the abstract you will find out that many of them discuss thrombotic events as well.
Author Response
We appreciate the time and effort you have invested in reviewing our work and are grateful for your contributions to its improvement. In the revised version, we have carefully addressed each of your points and incorporated the necessary changes to strengthen the manuscript. Below, we provide a detailed response to your comments, outlining the modifications made and clarifying any issues raised.
The authors have answered to many of my comments but have made no effort to improve the manuscript despite the fact that it was submitted to a high-impact journal. Although I can partially agree with some of your answers, the references remain too old. There are over 12000 papers on APL in MEDLINE and many recent papers on thrombotic events in APL. I found a paper identifying biochemical predictors of thrombotic events published this year. You can also compare the AIDA regimen to ATRA plus ATO in terms of thrombosis occurrence. If you would read other studies in full length and not only the abstract, you will find out that many of them discuss thrombotic events as well.
Response: Thank you for your suggestion. We initially considered the reference by Song et al. (2024) but the study analyses both arterial and venous thromboses (with only 5 VTE cases) and their predictors together. Since our work focuses exclusively on VTE, we decided to exclude the mentioned reference. However, considering your opinion and suggestion, we have now added a sentence about biochemical risk factors in the Introduction (Paragraphs 67-69) and included this reference. In our study, besides parameters (CBC, CTC, molecular parameters etc.) summarized in Table 1 and 2, we did not analyze the significance of biochemical parameters. In addition to the mentioned reference, we included another reference from 2023. (Hisada et al.), that explored predictors for bleeding and thrombosis in acute leukemia, including 29 APL patients. Although we did not performed literature search in structured and preplanned manner (like in systematic review) we did find a vigorous number of references during the manuscript preparation. However, we have tried to focus on those which are the most relevant to our specific topic in order not to obscure the valuable information in the long crowded text.
Regarding the comparison of AIDA and ATO-based regimens, we have added a few sentences in the Discussion section in Paragraphs 226-231. Few studies have examined thrombosis in ATO-based regimens, and the incidence is similar to ATRA-chemotherapy combination. Direct comparisons between these two modalities regarding VTE occurrence are lacking
Round 3
Reviewer 2 Report
Comments and Suggestions for Authors
The paper can be accepted